# Increase in Referrals of Children and Adolescents to the Psychiatric Emergency Room Is Evident Only in the Second Year of the COVID-19 Pandemic—Evaluating 9156 Visits from 2010 through 2021 in a Single Psychiatric Emergency Room

**DOI:** 10.3390/ijerph19158924

**Published:** 2022-07-22

**Authors:** Chen Dror, Nimrod Hertz-Palmor, Yael Yadan-Barzilai, Talia Saker, Maya Kritchmann-Lupo, Yuval Bloch

**Affiliations:** 1Shalvata Mental Health Center, Hod Hasharon 45100, Israel; yaelyadan1@gmail.com (Y.Y.-B.); taliasa2@clalit.org.il (T.S.); maiakr@clalit.org.il (M.K.-L.); yuvalbl@clalit.org.il (Y.B.); 2Sackler Faculty of Medicine, Tel Aviv University, Tel Aviv 69978, Israel; 3The Child and Adolescent Psychiatry Division, Sheba Medical Center, Edmund and Lily Safra’s Children Hospital, Ramat Gan 52621, Israel; 4School of Psychological Sciences, Tel Aviv University, Tel Aviv 69978, Israel

**Keywords:** COVID-19, psychiatric emergency room, pediatric mental health

## Abstract

**Introduction:** The COVID-19 pandemic affected the wellbeing of children and adolescents. The psychiatric emergency room (ER) is the hub of psychiatric emergencies and reflects clinically significant mental problems. Previous studies compared 2019 and 2020 and observed a decline in ER referrals. The current study focused on the continuous trend of referrals from 2010 to the end of 2021. **Method:** In our observational retrospective study, we procured data from 9156 child and adolescent referrals to our psychiatric ER. The comparison was made based on similar months of each year. **Results:** There was a significant positive trend in monthly referrals between 2010 and 2021, representing a similar increase in referrals per month in comparison to that month in the preceding year (*unstandardized β* = 4.21, 95% *CI* = 3.44 to 4.98, *p* < 0.0001). Between March 2020 and February 2021 (monthly visits = 72.5 + 16.6 [median = 79.5], annual referrals = 870), we observed no additive effect beyond this general trend after controlling for population growth. Conversely, between March and December 2021 (monthly referrals = 106.1 + 31.8 [median = 105.5], overall referrals = 1061) we observed a significant additive effect beyond the projected incline, as predicted by previous years (*β* = 21.61, 95% *CI* = 12.12 to 31.06, *p* < 0.0001). **Conclusions:** The first year of the COVID-19 pandemic was no different from the continuous decade long rise of referrals to the children and adolescents’ psychiatric ER. Conversely, the second year showed an additional incline beyond the general trend. The complexity in this rising need demands the awareness of clinicians and policy makers alike.

## 1. Introduction

The COVID-19 pandemic affected the lives of everyone immensely [1,2,3]. The continuous and challenging course of the pandemic is making the understanding of its effects on the mental health of children and adolescents extremely complex. Lockdowns, social distancing, lack of school and recreational routine, fear of being infected and fear of infecting others, the loss of loved ones, and economic and familial instabilities are all major stressors, affecting the mental health of children and adolescents [3,4]. Most studies that investigated the effect on children and adolescents’ mental health focused on “the waves” of mass outspread, i.e., restricted periods that are characterized by substantial increases in COVID-19 infections. Particularly, most studies to date focused on the first wave (March–April of 2020). When exploring the effects of COVID-19, the comparison was usually restricted to the years 2020 (the first year of COVID-19) and 2019 (the year before the outbreak) [5,6,7,8]. We consider this a limitation, as there is an increase in the use of pediatric mental health services in general over the years not related to the pandemic. For example, at the Shalvata Mental Health Center in Israel, the number of yearly referrals of pediatric patients to the psychiatric ER grew from 827 referrals in 2017 to 910 referrals in 2018, and to 1040 referrals in 2019. Thus, these effects can be masked when comparing 2019 to 2020 alone.

Community survey studies reported a rise in stress mainly during the first year of COVID-19, although not necessarily an incline in clinically relevant psychopathology [9]. In line with those surveys, lay publication reports from hotline services and community volunteer organizations present an increase in anxiety, depression, deliberate self-harm, and suicidal thoughts [10].

Beyond community surveys, there are studies that refer to clinical outcomes [11,12]. A central challenge of these observational studies is to identify covert mental illness, which is always partly related to difficulty in seeking help from mental health services. In a survey that included 2111 adolescents from the UK with a history of mental illness, 83% agreed that the pandemic worsened their mental health, and 26% said that they were no longer able to access mental health support [13].

Although there were reports on an incline in self-harm behaviors among adolescents, the number of patients presenting to ERs due to self-harm actually decreased in 2020, compared to 2019 [14].

It was assumed that emergency referrals to the emergency room (ER) would be less affected, as both psychiatric, as well as general medical ER facilities, continued to function in most countries. Most studies show a decline in psychiatric ER referrals for children and adolescents [5,15]. A partial explanation for this decline was the fact that, during lockdowns, there was a suspension of most academic, social, and recreational activities, which potentially alleviated mental stress in children and adolescents who suffer from separation anxiety, social anxiety, and school refusal [16]. In addition, patients suffering from bullying, academic difficulties, and other stressors may experience home schooling as a relief [17,18]. Moreover, there was a decline in the utilization of all emergency medical services for life-threatening medical conditions that are not related to COVID-19, which was the result of a general reduction in help-seeking behaviors during the first year of the pandemic [19,20].

However, during the second year of the pandemic, schools were mostly open, and thus all school related stressors described above started affecting children and adolescents again. Data regarding the burden that the COVID-19 pandemic had on child and adolescent mental health services beyond the first year of the pandemic is scarce. The trajectories of mental health stressors evolve over longer periods of time.

It is possible that some of the mental health consequences of a pandemic become evident only months after the stressors appear. For example, there is a delayed onset in 25% of post-traumatic stress disorder cases [21]. It is expected that symptoms and stress related disorders probably increase due to the exhaustion brought on by the prolonged stressors that are part of living with the pandemic.

In the current ecological and observational retrospective study, we aim to compare child and adolescent referrals to the psychiatric ER during the pandemic period (2020–2021) to the decade before (2010–2019) [22]. We examined whether trends in the number of ER referrals during the pre-pandemic decade continued or changed during the two years of the pandemic. We hypothesized that over time (comparing the first pandemic year 2020 to the second 2021) there would be a continuous incline in referrals to pediatric psychiatric ERs.

## 2. Method

### 2.1. Sample

The current study focused on minors, aged 6–18, who were referred to the psychiatric ER at Shalvata Mental Health Center between 1 January 2010 and 31 December 2021 in need of acute psychiatric care. No inclusion criteria were established other than visiting the ER during the relevant timeframe. A total of 9156 patients arrived within this 12-year time period. The Shalvata Mental Health Center serves a population of approximately 500,000 inhabitants in the center of Israel. Our sample represents the heterogeneous ethno-cultural population of Israel and thus probably allows for better generalization for different populations. It is important to note that there were differences in exposure to COVID-19 among different groups in our population. The Muslim minority and Ultraorthodox Jewish community showed more cases of infection, possibly at least partially due to less awareness of the dangers of the pandemic and more suspicion towards regulations set by the government [23].

### 2.2. Procedure

This was an ecological and observational retrospective study using data that were collected from the medical electronic files of patients. We focused on the monthly number of referrals for first-time patient referrals versus regularly treated patients’ referrals. Regularly treated patients were defined as patients who had therapy in our outpatient clinic within the past year. This study was approved by the institutional review board (IRBs) of Shalvata (0015-20-SHA).

### 2.3. Data Analysis

A mixed-effects linear regression was conducted to address spontaneous trends in the number of ER referrals (model 1) over the years (modeled as a fixed factor), with respect to the month-dependent fluctuations in the number of ER referrals (modeled as random effects). The choice to introduce months as a random factor was due to the seasonality of ER referrals [24], since they traditionally increase in certain months (in our center—referrals tend to increase during spring and winter and decrease during summer vacation). To address the additive effect of the COVID-19 years (2020–2021) on the increase in referrals beyond a spontaneous incline, we modeled the years 2020 and 2021 separately as dummy variables, and contrasted against the years 2010–2019 (modeled as the reference variable). COVID-19 was declared a pandemic in Israel in March 2020. Therefore, in all of our models we used ‘pandemic years’ instead of calendar years, meaning that we measured years from March to February of the following year, instead of measuring from January to December. As such, the year 2021 included 10 observations and not 12 since we were unable to obtain data from the year 2022 and modeled only the months March to December. We then repeated the same modeling method with the percentage of regularly treated patients (i.e., patients who had therapy in our outpatient clinic in the past year; model 2) from the total number of patient referrals. One exception from the first (ER referrals) model was that the data in model 2 were only available for the years 2017–2021, and not 2010–2021. Overall, we conducted two separate models and controlled the false discovery rate (FDR) with Benjamini and Hochberg’s FDR correction [25]. To control for population growth over the years, we collected data regarding population growth in the major cities covered by Shalvata Mental Health Center (data were obtained from the official site of the Israel Central Bureau of Statistics) and conducted a sensitivity analysis in which we introduced the overall population to the model as a random factor. We considered results to be statistically significant using the standard α < 0.05 chance for a type-I error, after correction. The analyses were conducted using the lmerTest package [26], and depicted using the ggplot2 package in R.

## 3. Results

The mean number of monthly ER referrals between 2010 and February 2020 was 59.39 ± 20.92 (median = 55). The number of annual ER referrals during these years was 712.7 ± 163.2 (median = 667), and overall, there were 7127 referrals between 2010 and February 2020. There was a significant positive trend in monthly referrals between 2010 and 2021, representing a continuous increase in referrals per month in comparison to the same month in the preceding year (*unstandardized β* = 4.21, 95% *CI* = 3.44 to 4.98, *p* < 0.0001). Between March 2020 and February 2021 (monthly referrals = 72.5 ± 16.6 [median = 79.5], annual visits = 870), we observed a significant decrease in the expected trend as projected by previous years (*β* = −10.06, 95% *CI* = −18.53 to −1.58, *p* = 0.022). However, this effect was eliminated after controlling for population growth (*p* = 0.14). Conversely, between March and December 2021 (monthly referrals = 106.1 ± 31.8 [median = 105.5], annual referrals [10 months only] = 1061) we observed a significant additive effect, beyond the projected incline as predicted by previous years (*β* = 21.61, 95% *CI* = 12.12 to 31.06, *p* < 0.0001) (Figure 1). This effect was maintained even after controlling for population growth (*p* = 0.009). Figure 1 and Figure 2 depict the average (Figure 1) and exact (Figure 2) number of monthly referrals over the decade.

We did not observe any trends in the monthly percentage of regularly treated patients versus first-time patients’ referrals (*β* = 0.00, 95% *CI* = −0.03 to 0.03, *p* = 0.99), and no additive effect in either 2020 (*p* = 0.15) or 2021 (*p* = 0.96).

## 4. Discussion

The present study attempts to examine clinically significant changes in the mental health emergencies of children and adolescents during the course of the COVID-19 pandemic.

We used referrals to the psychiatric ER as a reflection of these emergencies. We know from previous studies in our center, as well as others, that the main reasons for children and adolescents’ referrals to the psychiatric ER are suicidal acts, thoughts, and ideations, deliberate self-harm, violence related to mental states, and significant clinical deterioration [27].

Most studies of psychiatric ER referrals during the pandemic only compared pandemic years with the year or months before the pandemic hit [6,7,8,15]. In the current study, we show that this type of analysis can be misleading since, at least in our center, there was a continuous and significant increase over the years of children and adolescent psychiatric ER referrals. The rise in the mental health needs of children and adolescents is steadily increasing. The COVID-19 pandemic, and especially its consequences, increased public awareness to mental health needs [28,29].

This study substantiates the steadily increasing need for mental health services, specifically for children and adolescents’ psychiatric ER. Thus, professionals, policy makers, and the public should note that the increase in need is continuous and not just related to a “wave”, i.e., a brief period. In accord with our findings, a recent report on the state of children’s mental health services in the UK showed a continuous growing need of children and adolescents who require mental healthcare services, which is not met [30]. The UK report strengthens our view on the importance of studying ER referrals as a reflection of the condition of mental health services in general. They describe a growing number of children approaching ERs due to psychosocial problems or unmet safeguarding needs. The use of the ER for these needs implies that the underfunded social and ambulatory mental health services fail to serve the rising needs [30].

Only after controlling for the general increase in ER referrals can we relate to a possible increase in children and adolescents’ emergency psychiatric needs during the pandemic. This is relevant both to the understanding of the effects of the pandemic on psychopathology and to the planning of psychiatric services. Previous studies of psychiatric ER referrals focused on the “early waves” of the pandemic [5,6,7,8]. Generally, these studies reported a decline in referrals [5,6,7,8,15]. In the current study, while the referrals to the ER in the first pandemic year, 2020, were not different from the general trend, in 2021, there was a significant rise. During the second year of the pandemic, most schools reopened, and thus all school related stressors, such as bullying and academic difficulties that began to affect children and adolescents again, could be related to the incline in ER referrals during this year. Moreover, as a result of closures and the reopening of schools during the current pandemic, there is a concern about an increase in school attendance problems, including truancy and school withdrawal, which negatively impact children and adolescents’ mental health [31]. These described stressors, along with the complexity involved in access to mental health services, predict a continuous rise in children and adolescents’ psychiatric emergencies. That is why there is greater need for improved communication between child and adolescent mental health services and schools, as well as the promotion of effective ways of reaching out to the community in order to identify children at risk for mental distress. One way to make mental healthcare more accessible is telemedicine, which is cheap and available even within the limited resources of public pediatric care and is cost-effective [14]. A recent systemic review showed that older adolescents, females, and those previously diagnosed with mental disorders or with special education needs, were at most risk for deterioration during the pandemic [14]. It is necessary to better understand the reasons that put these groups at greater risk in order to customize their treatment. Further studies need to identify specific reasons for the reported increase in ER referrals. Our study emphasizes the need to make non-emergency psychiatric services more available and implies that improving mental health services should be highly prioritized.

## 5. Limitations

Our study had few limitations. Clinical data were collected from the medical files of the patients and were not assessed systematically using standardized rating scales or interviews. Additionally, important individual-level demographic data could not be obtained and adequately matched to the aggregated data due to technical limitations and ethical considerations of the medical filing system. Future studies should aim to explore factors such as age, gender, ethnicity, and psychiatric diagnosis. Our study relates to the changes in practice during the COVID-19 pandemic, but we could not assess psychiatric patients who did not seek mental care or used alternative options in the community. Another limitation is the fact that our study is a retrospective study, and as such, limits our ability to retrieve more data about the patients, such as information regarding patient complaints and diagnoses during their ER referrals, which could be relevant to better understanding the reasons for the results shown in the study. Moreover, the external validity of our study is limited by the fact that we reported data only from one ER in Israel.

## 6. Conclusions

When measuring the effects of the COVID-19 pandemic on the mental health among children and adolescents, it is highly important to consider whether current phenomena differ from existing trends. We demonstrated that the rise in ER referrals observed in 2021 was above and beyond the expected annual rise in previous years, which highlights the myriad increase in the seeking of mental healthcare during this complicated epoch. The complexity in this rising need demands the awareness of clinicians and policy makers alike. 

## Figures and Tables

**Figure 1 ijerph-19-08924-f001:**
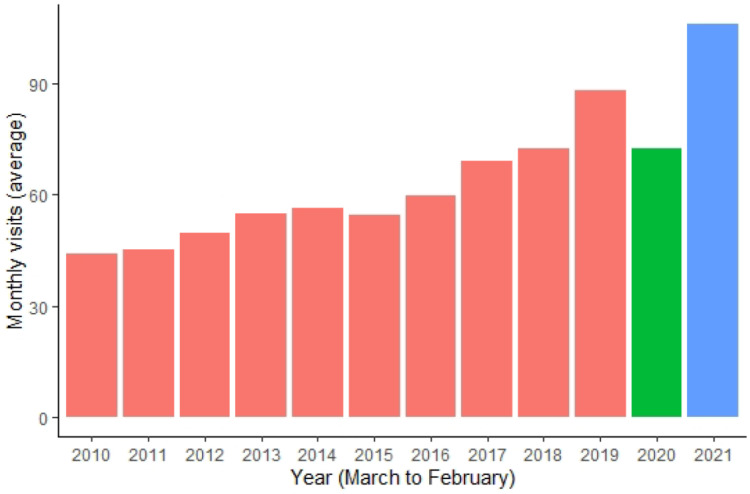
Average monthly ER referrals between the years 2010–2021. *X* axis values relate to “pandemic years”, i.e., referrals between March in the relevant year to February in the following year, expect for 2021 were data were obtained only for the months March to December.

**Figure 2 ijerph-19-08924-f002:**
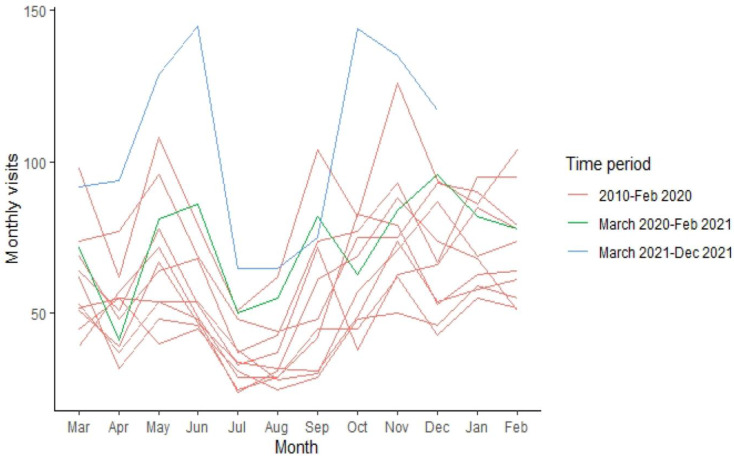
ER referrals between the years 2010–2021. Lines represent separate years.

## Data Availability

Data will be shared upon request.

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
