# Peer review of "Increase in Referrals of Children and Adolescents to the Psychiatric Emergency Room Is Evident Only in the Second Year of the COVID-19 Pandemic—Evaluating 9156 Visits from 2010 through 2021 in a Single Psychiatric Emergency Room"

_ijerph, 2022, doi:10.3390/ijerph19158924_

Round 1

Reviewer 1 Report

This paper uses data on child and adolescent visits to a psychiatric emergency department and mixed effects linear regression models to examine whether the COVID-19 pandemic is associated with increased visits. Such an increase could serve as a proxy indicator of increased stress and need for mental health services brought on by the pandemic. I list some suggestions below for the paper: 

--one question I had while reading the manuscript was what the unique contribution was. The brief literature review points out that previous observational studies showed psychiatric ER referrals for children and adolescents decreasing during the pandemic and provides some persuasive possibilities for why this might be the case. This leaves the question, however, of what the current study is going to illuminate for us. This requires a writing solution rather than a data solution, and in my opinion is an easy problem to solve. One possibility would be to rewrite this section slightly to more explicitly set those pieces up as “in competition” with pieces later in the paragraph that suggest increasing use of mental health services. That might allow the authors to set their contribution up as adjudicating among mixed evidence. Another possibility is emphasizing the comparison of pandemic data with the decade before; this might require demonstrating that previous literature did not make such comparisons. The discussion seems to suggest this is important, but it is an important hook to include earlier in the paper to contextualize the rest of the information. 

--the authors mention this very briefly, but it would be helpful for them to discuss what their findings mean for mental health in general given that they are looking at not just a clinical population, but a population that sought emergency care. They could use this limitation to speculate about how we should look across pandemic waves to assess other mental health outcomes. 

--I’m intrigued by the use of mixed-effects models, bur I think it would be beneficial if the authors gave a more thorough explanation of why they use these models and how they can show us things simpler models cannot. Put another way, it is probably not the best idea to let a reader wander around those ideas themselves rather than showing them what you want them to look at! 

--Since these kinds of models were being employed, I was also curious as to why the authors did not include potentially useful individual-level demographic covariates; they certainly introduce some good reasons to do so when describing their sample. Finally, perhaps I somehow downloaded the manuscript incorrectly, but in the manuscript I had access to, there were no tables. I found it difficult to assess how confident I felt about the findings without being able to consider them in more detail.

 --given the way the paper looks across time, I wondered if a data visualization might be a useful way to demonstrate findings.

Reviewer 2 Report

This paper describes an interesting public health topic related to mental health in adolescent. In particular the paper highlighted the continuos rising of need of mental health services in adolescents. Furthermore, the paper highlights that this trend was further incremented by COVID-19 pandemic.

The paper is well-written and scientifically sound. I would reccomend to further describe the indirect impact that COVID-19 pandemic had on other health services in the introduction section so as to show how covid-19 pandemic can affect health services in several ways.

Suggested references:

- Birkmeyer JD, Barnato A, Birkmeyer N, Bessler R, Skinner J. The Impact Of The COVID-19 Pandemic On Hospital Admissions In The United States. Health Aff (Millwood). 2020 Nov; 39(11):2010–2017.  https://doi.org/10.1377/hlthaff.2020.00980 Epub 2020 Sep 24. PMID: 32970495.

- Lastrucci V, Collini F, Forni S, D’Arienzo S, Di Fabrizio V, Buscemi P, et al. (2022) The indirect impact of COVID-19 pandemic on the utilization of the emergency medical services during the first pandemic wave: A system-wide study of Tuscany Region, Italy. PLoS ONE 17(7): e0264806. https:// doi.org/10.1371/journal.pone.0264806

Reviewer 3 Report

This report looks at children and adolescents' visits to our psychiatric ER and compares the data during the pandemic and the decade before. It is an exciting topic, but data collected via patients' medical files cannot tell much, especially regarding the factors related to the rise in the need for mental health services. The conclusion of this report shows the continuous rise of mental health service needs, not just related to the waves of the COVID-19 pandemic, which lacks academic innovation. Urbanization, digital life and many others result in worsened mental health situations, which is not new and not just in Israel. 

Round 2

Reviewer 1 Report

The authors have done a fine job of addressing reviewer concerns, and this version of the paper is much stronger in terms of outlining contributions.

Reviewer 3 Report

As one of the authors explained, this report reveals some facts that the state of mental health services is still in deepening crisis, as a response to the recent studies reporting the decline in approaching mental health services during the first year of the pandemic compering to 2019. I think it starts to make sense to me.